# Macro–Mesoscale Modeling of the Evolution of the Surface Roughness of the Al Metallization Layer of an IGBT Module during Power Cycling

**DOI:** 10.3390/ma16051936

**Published:** 2023-02-26

**Authors:** Tong An, Xueheng Zheng, Fei Qin, Yanwei Dai, Yanpeng Gong, Pei Chen

**Affiliations:** 1Institute of Electronics Packaging Technology and Reliability, Faculty of Materials and Manufacturing, Beijing University of Technology, Beijing 100124, China; 2Beijing Key Laboratory of Advanced Manufacturing Technology, Beijing University of Technology, Beijing 100124, China

**Keywords:** Al metallization layer, power cycling, surface roughness, macro–mesoscale modeling, crystal thermo–elasto–plasticity finite element

## Abstract

One of the main failure modes of an insulated-gate bipolar transistor (IGBT) module is the reconstruction of an aluminum (Al) metallization layer on the surface of the IGBT chip. In this study, experimental observations and numerical simulations were used to investigate the evolution of the surface morphology of this Al metallization layer during power cycling, and both internal and external factors affecting the surface roughness of the layer were analyzed. The results indicate that the microstructure of the Al metallization layer evolves during power cycling, where the initially flat surface gradually becomes uneven, such that the roughness varies significantly across the IGBT chip surface. The surface roughness depends on several factors, including the grain size, grain orientation, temperature, and stress. With regard to the internal factors, reducing the grain size or orientation differences between neighboring grains can effectively decrease the surface roughness. With regard to the external factors, the reasonable design of the process parameters, a reduction in the stress concentration and temperature hotspots, and preventing large local deformation can also reduce the surface roughness.

## 1. Introduction

Insulated-gate bipolar transistors (IGBTs) have been extensively applied in rail traffic, automotives, high-speed trains, and wind-power generation for electric energy control and conversion. To facilitate wire bonding for the interconnection of chips and substrates, an aluminum (Al) metallization layer is sputtered on the surface of an IGBT or a freewheel diode (FWD) chip. The surface morphology of the Al metallization layer evolves during power cycling [1,2]. The corresponding reduction in the effective cross-sectional area of the layer significantly increases the resistance of the entire Al metallization layer, causing the voltage between the IGBT collector and emitter to rise and hotspots to appear in some regions. The effective connection area of wire bonding is also decreased, and the peeling or cracking of the Al bonding wires is accelerated, resulting in the failure of the IGBT module [3,4]. Therefore, studying the evolution mechanism of the surface morphology of the Al metallization layer is crucial for optimizing and improving the operational reliability of IGBT modules.

Both internal and external factors affect the surface roughness of the Al metallization layer. The main external factors are the temperature and stress. The junction temperature of an IGBT chip rises during power cycling due to the power loss of the IGBT module and varies with the output power [5,6]. Cyclical temperature fluctuations lead to the repeated thermal expansion and contraction of the material layers of the IGBT module [7,8]. The large mismatch between the coefficients of thermal expansion (CTEs) of the silicon (Si) chip and Al metallization layer repeatedly subjects the layer to thermal stress [9,10]. The diffusion of Al atoms along grain boundaries to release this stress creates dislocations and, thus, leads to the plastic deformation of the Al grains. Finally, extruded Al grains protrude near grain boundaries, changing the surface morphology [11]. As plastic deformation induced by thermal stress is the main reason for the degradation of the Al metallization layer [12], changes in the junction temperature can significantly affect the surface roughness of the layer. Smet et al. [13] demonstrated that the surface roughness of the Al metallization layer increases with the junction temperature for a fixed number of power cycles. In addition, the Al metallization layer in the central chip area with a high junction temperature is rougher than at the edge [1,3].

Both the internal factors of the grain size and grain orientation affect surface roughness evolution. Raabe et al. [14] conducted a finite element (FE) analysis on the unidirectional compressive deformation process of single-crystal copper (Cu) to study the effect of factors such as the grain shape, initial orientation difference, and friction coefficient on the plastic slip in the crystal. The grain shape and initial orientation difference in polycrystalline materials were found to significantly impact plastic deformation. Liu et al. [15] combined a crystal plastic constitutive model with the actual microstructure of polycrystalline materials to develop a crystal plasticity FE model, and studied the evolution of the surface roughness. The grain size and texture were found to affect the evolution of the surface roughness. Feng et al. [16] used a macro–mesoscale modeling method and a two-dimensional (2D) crystal plasticity FE model to study the surface roughness of a material during deformation processing, and analyzed the effect of the grain size on the surface roughness.

In this study, the microstructural evolution of an Al metallization layer of an IGBT module during power cycling was observed. Multilevel submodeling technology, Voronoi tessellation, and crystal thermo–elasto–plasticity FE models were used to conduct qualitative macro–mesoscale simulations to analyze the effect of factors such as the thermal stress, grain orientation, and grain size on the evolution of the surface roughness of the Al metallization layer.

## 2. Experimental Process

### 2.1. Experimental Sample of an IGBT Module

Figure 1 shows the test sample, a 1200 V/450 A half-bridge IGBT module. The upper and lower bridge arms of the module are composed of two groups of IGBT chips connected in parallel. Each group contains three IGBT chips in an antiparallel configuration to an FWD chip. There is a total of six IGBT chips and six FWD chips welded on three direct copper-bonded (DCB) substrates. The emitter on the surface of the IGBT chip and the anode of the FWD are electrically connected to the upper Cu layer of the DCB substrate through Al bonding wires. To facilitate wire bonding, a 4 μm thick Al metallization layer is sputtered on the chip surface.

Power cycling tests are performed on four IGBT modules, A, B, C, and D, using different numbers of power cycles. The tests are performed on three IGBT chips of the upper half bridge arm of each module. The chips on each module are labeled as “module number-chip number”, e.g., A-I corresponds to Chip I on Module A. The power cycling test is conducted on Module D until the module fails, corresponding to a total of 193 k cycles. The other three modules, A, B, and C, are subjected to 0, 100 k, and 164 k cycles, respectively, i.e., 0%, 50%, and 85% of the failure life (in terms of the number of power cycles).

### 2.2. Power Cycling Test

Figure 2 shows the power cycling test platform, which consists of a main power source, test power source, signal generator, drive protection circuit, data acquisition system, and water cooling system. Figure 3 shows the circuit diagram for direct current (DC) power cycling. Two IGBT modules are applied here; one is the device under test (DUT), and the other one is the control switch. After the test begins, a constant driving voltage of 15 V is applied to the gate of the DUT to ensure the device remains conductive. The control switch is turned on to connect the main circuit and supply a constant load current (*I*_load_) of 400 A to the DUT. The power loss causes the DUT junction temperature to rise rapidly. Then, the control switch is turned off to cut off the main circuit, and the cooling system is turned on to ensure that the DUT cools down rapidly. A control strategy of a fixed turn-on and turn-off time (*t*_on_ = 2 s and *t*_off_ = 2 s) is employed during the test. A 4 s cycle is used, and the temperature of the water cooling system is set to 45 °C. Additionally, to estimate the junction temperature of the DUT, a constant measurement current (*I*_measure_) of 100 mA was applied. The on-state collector–emitter voltage *V*_ce-on_ at 100 mA was measured when the load current was disconnected.

### 2.3. Experimental Observation of the Evolution of the Microstructure of the Al Metallization Layer

After the power cycling test is completed, the evolution of the microstructure of the Al metallization layer on the surfaces of the chips of the four IGBT modules is observed. Figure 4 shows the steps used to characterize the Al metallization layer. First, the IGBT modules are processed by removing the housings, lead terminals, and silica gel [17]. Second, the surface morphology of the Al metallization layer is observed using scanning electron microscopy (SEM). The surface roughness is measured before and after the power cycling test using atomic force microscopy (AFM). Third, the Al metallization layer is ion-thinned using a focused ion beam (FIB) for observation of the layer cross-section with FIB-SEM. Fourth, electron backscatter diffraction (EBSD) is employed to analyze the texture of the FIB-etched region.

To measure the surface roughness of the Al metallization layer, tests are performed at three representative positions on each IGBT chip, as shown in Figure 4: P1—the central chip area, P2—near the bond wire heel, and P3—at the edge of the chip. A 40 μm × 40 μm area is selected at each position to obtain a three-dimensional (3D) morphology file of the Al metallization layer surface through AFM. Then, the 3D morphology file is imported into Gwyddion 2.45 software to draw a 3D morphological map. The arithmetic mean surface roughness *S*_a_ of the entire 3D morphological map is calculated, i.e., the mean value of the absolute deviation *Z* of each point on the 3D morphological map relative to the reference plane:(1)$$S_{a}=\frac{1}{A}\iint_{A}\left|Z\left(x,y\right)\right|dxdy$$
where *A* is the number of sampling nodes and *Z*(*x*, *y*) is the deviation of the sampling node height *Z_i_*(*x*, *y*) relative to the middle plane *Z*_0_(*x*, *y*), i.e., *Z*(*x*, *y*) = *Z_i_*(*x*, *y*) − *Z*_0_(*x*, *y*).

## 3. Numerical Analysis of the Surface Roughness of the Al Metallization Layer of the IGBT Module

A macro–mesoscale model for the Al metallization layer of the IGBT module is formulated to investigate the changes in *S*_a_. First, a thermal–electrical–structural FE analysis is conducted on the IGBT modules under the power cycling test conditions. Then, submodeling technology is used to impose the boundary conditions on a grain-level model of the Al metallization layer at specific positions. A geometric model for the Al grains is built using Voronoi tessellation. A crystal thermo–elasto–plasticity constitutive equation is numerically implemented by incorporating a user-defined subroutine UMAT into the finite element code ABAQUS to simulate the evolution of the surface roughness of the Al metallization layer.

### 3.1. Submodel for the IGBT Module

To elucidate the evolution mechanism of the surface roughness of the Al metallization layer and determine the corresponding influence factors, submodeling technology and Voronoi tessellation are used to create a geometrical model, which is shown in Figure 5. In the global model, an FE model for a single-chip IGBT module is created. Figure 5a shows this FE model consisting of an Al bonding wire, Al metallization layer, IGBT chip, chip solder, upper Cu layer, ceramic layer, lower Cu layer, DCB solder, and Cu substrate. The model dimensions are listed in Table 1. The model includes a total of nine Al bonding wires of the same size placed approximately 1 mm apart.

A three-level submodel is formulated to impose the boundary conditions more efficiently. Three first-level submodels are formulated to comparatively analyze the roughness at different positions on the chip surface. The first-level submodel at the chip center (P1) is selected from the fifth bonding wire of the global model and has dimensions of 19.5 mm × 1 mm × 6.424 mm, including the Al bonding wire and the lower structures of each layer, totaling 71,960 elements. The first-level submodel at the bonding wire heel (P2) is selected from the middle of the seventh and eighth bonding wires of the global model and has dimensions of 19.5 mm × 1 mm × 4.604 mm, excluding the bonding wires and only including the Al metallization and lower structures of each layer, totaling 70,205 elements. The first-level submodel at the edge of the chip (P3) is selected from the ninth bonding wire of the global model and has dimensions of 19.5 mm × 1 mm × 6.424 mm, including the Al bonding wire and lower structures of each layer, totaling 71,960 elements.

The second-level submodel is selected from the central part of the first-level submodel and has dimensions of 0.32 mm × 0.32 mm × 0.154 mm, including the Al metallization layer and the chip layer underneath, totaling 98,304 elements.

The third-level submodel is selected from the central part of the second-level submodel and has dimensions of 0.03 mm × 0.03 mm × 0.004 mm, including only the Al metallization layer, which is divided into polycrystalline geometric shapes. There are 142,519 elements in total. The Voronoi tessellation algorithm [18,19] is used to create a geometric model of the polycrystalline structure of the Al metallization layer. A Voronoi diagram is generally constructed using a set *P* of *n* discrete points on a two-dimensional plane, i.e., *p*_1_, *p*_2_..., *p*_n_ ∈ *P*, where *V*(*p_i_*) is defined to satisfy the following equation:(2)$$V\left(p_{i}\right)=\left\{\;x\in P\;|\;d\;\left(x,p_{i}\right)\leq d\;\left(x,p_{j}\right)\;,\;\forall \;j\neq i\;\right\}$$
where *d*(*x, pi*) is the Euclidean distance between points *x* and *p_i_*. Then, *V*(*p_i_*) is the Voronoi polygon of point *p_i_*. The randomly generated discrete point *p*_i_ is called the generator of the Voronoi diagram, and each Voronoi polygon obtained represents a grain.

The coordinate values of the Voronoi diagram generated using MATLAB are saved as a data file in a prescribed order. The data file of the vertices is read using the Python scripting language, followed by connecting the vertices in sequence. Finally, the interface with Python in ABAQUS/CAE is used to generate a polyhedron as a geometric model of the polycrystalline structure.

### 3.2. Crystal Thermo–Elasto–Plasticity Constitutive Equation

#### 3.2.1. Kinematics

The total deformation gradient of a crystalline material ***F*** can be expressed using the single-crystal plasticity model proposed by Asaro and Rice [20] and considering the deformation caused by a temperature change [21]:(3)$$F=F^{e}\cdot F^{p}\cdot F^{\theta }$$
where ***F***^e^ is the crystal elastic deformation gradient, ***F****^θ^* is the thermal deformation gradient caused by free thermal expansion, and ***F***^p^ is the crystal plastic deformation gradient.

Assuming the isotropic expansion of the material, ***F****^θ^* can be expressed as follows [22]:(4)$$F^{\theta }=\left(1+q\cdot \Delta \theta \right)I$$
where *q* represents the CTE, Δ*θ* denotes the temperature change, and ***I*** denotes a second-order tensor.

The plastic component of the deformation gradient ***F***^p^ can be expressed as follows [23]:(5)$$F^{p}=I+\gamma^{\left(\alpha \right)}\sum_{\alpha =1}^{N}s_{0}{}^{\left(\alpha \right)}\cdot m_{0}{{}^{\left(\alpha \right)}}^{{}^{T}}$$
where *γ*^(*α*)^ is the shear strain in the *α*th slip system. Lattice distortion occurs when a crystal transforms from an intermediate plastic configuration to an instant configuration under a deformation gradient ***F***^e^. Then, the vector in the slip direction and the normal vector on the slip plane of the αth slip system under the instant configuration are denoted by ***s***^(α)^ and ***m***^(α)^, respectively:(6)$$s^{\left(\alpha \right)}=F^{e}\cdot s_{0}{}^{\left(\alpha \right)}$$
(7)$$m^{\left(\alpha \right)}={\left(F^{e}{}^{{}^{T}}\right)}^{-1}\cdot m_{0}{}^{\left(\alpha \right)}$$

Let ***l*** be the velocity gradient, then:(8)$$l=\dot{F}\cdot {\left(F\right)}^{-1}$$

The velocity gradient ***l*** is decomposed as follows:(9)$$l=\overset{\cdot }{\left(F^{e}\cdot F^{p}\cdot F^{\theta }\right)}\cdot {\left(F^{e}\cdot F^{p}\cdot F^{\theta }\right)}^{-1}=l^{e}+l^{p}+l^{\theta }$$
where ***l***^e^, ***l***^p^, and ***l****^θ^* are the elastic, plastic, and thermal contributions to the velocity gradient, respectively.

Substituting Equation (5) into ***l***^p^, and combining Equations (6) and (7) yields:(10)$$l^{p}={\dot{\gamma }}^{\left(\alpha \right)}\sum_{\alpha =1}^{N}s^{\left(\alpha \right)}\cdot m^{\left(\alpha \right)}{}^{{}^{T}}$$

The velocity gradient ***l*** is decomposed into a symmetric part (the deformation rate tensor ***d***) and an antisymmetric part (the rotation rate tensor ***w***):(11)l=d+w

These two tensors can be decomposed into elastic, thermal, and plastic components:(12)$$d=d^{e}+d^{p}+d^{\theta }$$
(13)$$w=w^{e}+w^{p}+w^{\theta }$$

Equations (11)–(13) can be used to express ***d****^θ^* and ***w****^θ^* as follows:(14)$$d^{\theta }=\frac{1}{2}\left[l^{\theta }+{\left(l^{\theta }\right)}^{T}\right]$$
(15)$$w^{\theta }=\frac{1}{2}\left[l^{\theta }-{\left(l^{\theta }\right)}^{T}\right]$$

Equations (10)–(13) can be used to express ***d***^p^ and ***w***^p^ as follows:(16)$$d^{p}=\frac{1}{2}\left[l^{p}+{\left(l^{p}\right)}^{T}\right]=\sum_{\alpha =1}^{N}P^{\left(\alpha \right)}{\dot{\gamma }}^{\left(\alpha \right)}$$
(17)$$w^{p}=\frac{1}{2}\left[l^{p}-{\left(l^{p}\right)}^{T}\right]=\sum_{\alpha =1}^{N}R^{\left(\alpha \right)}{\dot{\gamma }}^{\left(\alpha \right)}$$

The symmetric and antisymmetric parts of the slip system are expressed as follows:(18)$$P^{\left(\alpha \right)}=\frac{1}{2}\left(s^{\left(\alpha \right)}\cdot m^{\left(\alpha \right)}{}^{{}^{T}}+m^{\left(\alpha \right)}\cdot s^{\left(\alpha \right)}{}^{{}^{T}}\right)$$
(19)$$R^{\left(\alpha \right)}=\frac{1}{2}\left(s^{\left(\alpha \right)}\cdot m^{\left(\alpha \right)}{}^{{}^{T}}-m^{\left(\alpha \right)}\cdot s^{\left(\alpha \right)}{}^{{}^{T}}\right)$$

#### 3.2.2. Rate-Dependent Hardening Model

An elastic constitutive relation considering the thermal stress term can be written as follows [24]:(20)$$\overset{\nabla }{\tau^{*}}=C\left(\theta \right):\left(d^{e}+d^{\theta }\right)$$
where $\overset{\nabla }{\tau^{*}}$ is the Jaumann objective stress rate of the Kirchhoff stress tensor ***τ*** with the intermediate temperature configuration as the reference condition, and ***C***(*θ*) is the fourth-order elastic tensor. The equation for $\overset{\nabla }{\tau^{*}}$ is:(21)$$\overset{\nabla }{\tau^{*}}=\dot{\tau }-\left(w^{e}+w^{\theta }\right)\cdot \tau +\tau \cdot \left(w^{e}+w^{\theta }\right)$$

The Jaumann objective stress rate of the Kirchhoff stress tensor ***τ*** with the initial configuration as the reference condition is:(22)$$\overset{\nabla }{\tau }=\dot{\tau }-w\cdot \tau +\tau \cdot w$$

Equations (12) and (13) for the rotation rate tensor can be combined with Equations (20)–(22) to yield:(23)$$\overset{\nabla }{\tau }=C\left(\theta \right):\left(d-d^{p}\right)-w^{p}\cdot \tau +\tau \cdot w^{p}$$

Substituting Equations (16)–(19) into Equation (22) yields:(24)$$\overset{\nabla }{\tau }=C\left(\theta \right):d-\sum_{\alpha =1}^{N}\left(C\left(\theta \right):P^{\left(\alpha \right)}-R^{\left(\alpha \right)}\tau +\tau R^{\left(\alpha \right)}\right){\dot{\gamma }}^{\left(\alpha \right)}$$

The relationship between the shear strain rate of the slip system and the decomposed shear stress is [25]:(25)$${\dot{\gamma }}^{\left(\alpha \right)}={\dot{\gamma }}^{\left(0\right)}{\left(\frac{\tau^{\left(\alpha \right)}}{g^{\left(\alpha \right)}}\right)}^{n}\mathrm{sgn}\left(\frac{\tau^{\left(\alpha \right)}}{g^{\left(\alpha \right)}}\right)$$
where *τ*^(*α*)^ is the decomposed shear stress of the single crystal in the *α*th slip system, *g*^(α)^ is the critical resolved shear stress, ${\dot{\gamma }}^{\left(0\right)}$ is the reference strain rate at *τ*^(α)^ = *g*^(*α*)^, *n* is the sensitivity exponent, and sgn is the sign function.

The modified Voce-type model is used to calculate the critical resolved shear stress rate ${\dot{g}}^{\left(\alpha \right)}$ [26]:(26)$${\dot{g}}^{\left(\alpha \right)}=\sum_{\beta }h_{\alpha \beta }\cdot {\dot{\gamma }}^{\left(\beta \right)}$$
where *h_αβ_* is the self-hardening modulus:(27)$$h_{\alpha \beta }=\left\{ \begin{array}{l} h(\gamma )=h_{0}\cdot {\mathrm{sech}}^{2}\left|\frac{h_{0}\cdot \gamma }{\tau_{s}-\tau_{0}}\right|\;\left(\alpha =\beta \right)\\ qh_{\alpha \alpha }\;\left(\alpha \neq \beta \right) \end{array}\right.$$
where *h*_0_ is the initial hardening modulus, *τ*_0_ is the initial critical resolved shear stress, *τ_s_* is the flow saturation stress, and *q* is the hardening ratio.

The crystal thermo–elasto–plasticity constitutive equation presented above is used to modify the crystal plastic subroutine UMAT proposed by Huang [27]. The effect of temperature on the fourth-order elastic tensor ***C***(*θ*) is introduced by thermal expansion, and the temperature deformation gradient ***F****^θ^* is introduced into the process of crystal elastoplastic deformation. The N-R iterative method is used to calculate the established implicit integral scheme and modify the stress affected by the temperature. The total stress at each moment is the sum of the calculated crystal plastic stress *τ*^p^ and calculated supplementary thermal stress *τ^θ^*.

The Al metallization layer has a face-centered cubic structure and, thus, 12 slip systems. Therefore, among the anisotropic elastic constants of the crystalline material, it is only necessary to set C11, C12, C44, the hardening ratio, and other related parameters. Table 2 shows the material parameters [28,29]. In the FE analysis of the global model, the Al metallization layer and Al bonding wire are considered elastoplastic materials with a yield strength of 30 MPa. The resistivity *y* of the IGBT chip depends on the temperature *x*, as y=0.0248x+4.5139 [30]. The other material parameters are listed in Table 3 [31].

### 3.3. Thermal–Electrical–Structural FE Analysis of the Global Model and Submodel

A thermal–electrical–structural FE analysis is conducted on the global model [34]. First, a thermal–electrical FE analysis is carried out to determine the temperature profile of the global model. Eight-node linear coupled thermal–electrical elements (DC3D8E) are used in the model. Then, the temperature profile is used as the load in a thermal–structural FE analysis to determine the stress profile of the global model. Eight-node linear brick elements with reduced integration (C3D8R) are used in the thermal–structural analysis. A total of 257,000 elements are used in the global model.

Figure 5 shows the thermal–electrical FE analysis load and boundary conditions of the global model. The rightmost end of the upper Cu layer 1 is used as the current inflow end at which the surface current load is applied. The leftmost end of the upper Cu layer 2 is set to be the “0” potential point as the boundary condition. To maintain consistency with the loading condition for the power cycling test, the current is 400 A, and the turn-on and turn-off times are *t*_on_ = 2 s and *t*_off_ = 2 s, respectively. The convective heat transfer coefficient of the lower surface of the Cu substrate is set to 2 W/(mm^2^·K) to represent the forced convective heat loss between the bottom of the module and cooling water. The ambient temperature is set to 23 °C through a predefined field. Then, the temperature profile obtained from the thermal–electrical FE analysis is applied to the model as a load to carry out a thermal–structural FE analysis of the global model. The lower surface of the Cu substrate is treated as a fixed constraint to prevent rigid body displacement. The simulation is run for 500 cycles for a total of 2000 s.

For the thermal–electrical–structural submodel, the analysis of each level is divided into heat transfer and thermal analyses. The same meshes are used for both analyses. For example, in the first-level submodel analysis, the heat transfer analysis is first performed to transfer the temperature field in the global model to the first-level submodel. Then, the thermal analysis is performed to assign the temperature field results from the heat transfer analysis to the first-level submodel, and transfer the displacement and stress fields determined using the global model to the first-level submodel for the thermal–structural analysis to determine the stress field. The same process is used to transfer fields from the first-level model to the second-level model to the third-level model.

## 4. Results and Discussion

### 4.1. Experimental Observation of the Evolution of the Surface Morphology of the Al Metallization Layer during Power Cycling

Figure 6 shows an SEM image of the surface morphology of the Al metallization layer at the center of the IGBT chip (P1) during power cycling. Under the initial conditions, the Al grains are arranged uniformly, and the Al metallization layer exhibits a planar surface morphology. The transistor cells are evenly arrayed and clearly visible (Figure 6a). After 193 k power cycles, the surface of the Al metallization layer becomes uneven, with long strips of folds, and the transistor cells are completely blocked and cannot be discerned (Figure 6b). These results show that the Al metallization layer becomes increasingly rough as the number of power cycles increases.

Figure 7 shows a 3D morphological map of the surface of the Al metallization layer at three different test positions on the IGBT chip and the measured *S*_a_ after different numbers of power cycles. The surface of the Al metallization layer becomes increasingly rough as the number of power cycles increases. Under the initial conditions, the Al metallization surface is even, and *S*_a_ at P1 is 0.064 μm. After 100 k power cycles, the surface morphology of the layer becomes uneven, and the roughness increases sharply. *S*_a_ at P1 increases more than sevenfold to 0.464 μm. After 164 k power cycles, there are more hillocks and microvalleys on the layer surface. *S*_a_ at P1 increases slightly by 18.10% to 0.548 μm. When the number of power cycles reaches 193 k, the hillocks on the layer surface gradually aggregate. Thus, the hillocks decrease in number and increase in size. *S*_a_ at P1 increases by 20.62% to 0.661 μm.

*S*_a_ increases with the number of power cycles at all test positions, but the extent of the increase depends on the position. Under the initial conditions, the *S*_a_ values at the three different positions (P1, P2, and P3) are essentially the same. *S*_a_ increases sharply with the number of power cycles, where *S*_a_ is largest at P1, followed by P2 and P3. That is, power cycling causes the Al metallization layer to degrade most severely at the IGBT chip center, where the surface is rougher than that at other positions.

### 4.2. External Factor Affecting the Surface Roughness of the Al Metallization Layer—Temperature

To study the effect of the temperature load on the evolution of the surface morphology of the Al metallization layer, three first-level submodels are established at the three test positions described in Section 3.1: P1—the central chip area, P2—near the bond wire heel, and P3—at the chip edge. All the third-level submodels used have exactly the same grain size, shape, and crystal orientation to exclude the influence of other factors.

Figure 8, Figure 9 and Figure 10 show the calculated strain and surface morphology fields of the Al metallization layer at three positions on the IGBT chip surface after 200 s, 1200 s, and 2000 s of power cycling. Figure 11 shows the variation in *S*_a_ with the number of power cycles determined by a statistical calculation of the normal plastic deformation of the surface. As the number of power cycles increases, the surface of the Al metallization layer at the three positions gradually changes from flat to uneven, and *S*_a_ clearly increases. For example, *S*_a_ at P1 is 5.29 × 10^−3^ μm after 200 s of power cycling. Increasing the cycling time to 2000 s causes *S*_a_ to increase nearly 19-fold to 0.100 μm.

The surface morphology of the Al metallization layer changes at all three positions but at different rates. After 200 s of power cycling, *S*_a_ is 5.29 × 10^−3^ μm, 4.08 × 10^−3^ μm, and 3.85 × 10^−3^ μm at P1, P2, and P3, respectively. The overall *S*_a_ is very low. The difference in *S*_a_ between P1 and P3 is 27.22%. After 1200 s of power cycling, *S*_a_ becomes 0.0388 μm, 0.0235 μm, and 0.0164 μm at P1, P2, and P3, respectively. The difference in *S*_a_ between P1 and P3 is 57.73%, which is a clear increase over that determined after 200 s. When the cycling time reaches 2000 s, *S*_a_ is 0.100 μm, 0.0531 μm, and 0.0323 μm at P1, P2, and P3, respectively. The difference in *S*_a_ between P1 and P3 rises to 67.70%. Thus, after power cycling, the highest increase in *S*_a_ occurs at P1, followed by P2 and P3. The difference in *S*_a_ between the positions increases with the number of power cycles. This result is consistent with the observed distribution of the surface roughness at the three test positions presented in the preceding section.

The variation in *S*_a_ with position mainly results from the uneven surface temperature of the IGBT chip during power cycling. Figure 12 shows the calculated temperatures at the three test positions during power cycling. During the turn-on stage of a cycle, the temperatures at the three positions continuously increase and peak at the moment the current is cut off. The maximum temperatures at P1, P2, and P3 are 157.87 °C, 153.77 °C, and 142.40 °C, respectively, where the maximum temperature at P1 is significantly higher than that at the edge. The nonuniform temperature distribution over the chip surface causes the thermal stress of the Al metallization layer to vary with position. The change in the surface morphology of the layer is mainly caused by plastic deformation resulting from thermal stress. Thus, an uneven distribution of *S*_a_ can be observed on the surface of the IGBT chip. In particular, *S*_a_ is largest at P1, where the temperature is highest, followed by P2 and P3.

Figure 12 shows that during each power cycle, the strain and the calculated *S*_a_ at the surface of the Al metallization layer fluctuate with the temperature. We consider P1 during 1196–1200 s of power cycling as an example. *S*_a_ is 0.0338 μm at 1196 s. From 1196 s to 1197 s, the temperature at P1 rises rapidly, the Al metallization layer expands, the strain increases continuously, and *S*_a_ at this moment is 0.0392 μm. As the temperature rises continuously up to 1198 s, the layer deforms more severely, but at a lower rate. At 1198 s, the temperature, strain, and *S*_a_ (0.0432 μm) all reach their maximum values. When the electric current is turned off, the temperature begins to fall, the layer starts to contract, and the elastic strain created during the temperature rise of the layer decreases. At the final moment of the cooling stage, i.e., at 1200 s, almost all the deformation of the layer is irrecoverable plastic deformation, and *S*_a_ has decreased to 0.0340 μm. Thus, *S*_a_ increases by 0.0002 μm, from 0.0338 μm to 0.0340 μm, during this cycle. When the power cycling time reaches 2000 s, *S*_a_ increases by 0.001 μm, from 0.0912 μm to 0.0922 μm, during one cycle. These results show that *S*_a_ increases during one power cycle, and the growth in *S*_a_ increases with the number of power cycles. This result is obtained because the strain distribution is essentially uniform when yielding begins. The small strain between grains decreases the shear stress of some grains with a small orientation difference to below the critical shear stress, which is insufficient to initiate slip in the system. Thus, there is little slip at the grain boundaries. As the strain grows, more slip systems are initiated, which increases the compatibility between deformations at grain boundaries. Then, more grains deform inside or at the grain boundary, rapidly increasing *S*_a_.

This behavior indicates that the temperature has a significant influence on *S*_a_ during power cycling. The higher the temperature, the more severely the surface morphology is degraded by power cycling. *S*_a_ is largest at P1, followed by P2 and P3; that is, the smallest change in the surface morphology and the lowest *S*_a_ occur at P3. In addition, the temperature exhibits cyclical fluctuations during power cycling. The deformation of the Al metallization layer also varies with the temperature during a cycle. As the number of power cycles increases, both the extent of plastic deformation and the rate of change in *S*_a_ increase.

### 4.3. Internal Factor Affecting the Surface Roughness of the Al Metallization Layer—Grain Orientation

To study the effect of the grain orientation on the evolution of *S*_a_, the same first- and second-level submodels located at P1 are selected to establish three third-level submodels with the same grain sizes and shapes. The average grain size is 4 μm, and the average orientation differences are 15°, 45°, and 90°.

Figure 13 shows the strain and surface morphology of the Al metallization layer calculated using the three models after 2000 s of power cycling. Figure 14 shows how *S*_a_ obtained using the three models with mean grain orientation differences of 15°, 45°, and 90° changes with the cycling time. Figure 13a shows that for a mean orientation difference between neighboring grains of 15°, two very large peaks in the height appear, which affects the overall average *S*_a_. A small grain orientation difference results in interconnections between some grains, such that grains with the same orientation evolve into a larger grain and slip between grains decreases considerably to reduce *S*_a_. However, the formation of abnormally large grains results in the transfer of the stress and strain to the boundary of the whole grain model along with the grain boundaries, and the strain in other regions decreases accordingly. The abnormally large grains undergo further deformation, resulting in severe strain localization and an extremely large peak in the surface roughness. This phenomenon is quite rare in the model with a large grain orientation difference.

As the simulation progresses, the surface roughness ratio of the model with an average crystal orientation difference of 90° decreases compared with other models. This indicates that the smaller the mean orientation difference, the larger the critical shear stress required for the crystal to slip, and the more difficult it is for the crystal to slip. However, with an increase in intercrystal stress and strain, the starting slip system increases gradually, and the surface deformation rate also increases.

Removing the peak in the surface roughness for the three models with mean grain orientation differences of 15°, 45°, and 90° results in a *S*_a_ of 0.068 μm, 0.100 μm, and 0.238 μm, respectively, after 2000 s of power cycling. That is, the *S*_a_ increases with the grain orientation difference. The reason for this result is that as the orientation difference between adjacent grains increases, the deviation in the deformation between grains increases, resulting in a rougher deformed surface. The *S*_a_ of the model with a grain orientation difference of 90° is 137% higher than that of the model with a grain orientation difference of 45°. This result indicates that the grain orientation has a considerable influence on the surface roughness of the Al metallization layer.

Figure 15 presents the experimentally observed grain orientation for the profile of the Al metallization layer in the chip center in the initial state and after 100 k cycles. The profile of the Al metallization layer in the initial state has a strong (111) crystal face texture, and each columnar crystal maintains the same out-of-plane (111) orientation from top to bottom. During the sputtering of the Al metallization layer, crystals with the (111) orientation grow first, such that the layer exhibits a distinct (111) crystal face texture at the initial stage. As the number of power cycles increases, the (111) orientation of the texture at the initial state is almost indiscernible, the texture changes to the (311) orientation, and the grain orientation difference increases dramatically. During the process of power cycling, the evolution of the surface morphology also accelerates as the grain orientation difference continues to increase.

### 4.4. Internal Factor Affecting the Surface Roughness of the Al Metallization Layer—Grain Size

To study the effect of the grain size on the evolution of the surface roughness, the same first- and second-level submodels at the chip center are used to establish three third-level submodels with grain sizes of 4 μm, 5 μm, and 6 μm, as is shown in Figure 16. The corresponding numbers of grains are 107, 55, and 32, respectively. The grain orientations in all the models are randomly distributed.

Figure 17 shows the strain and surface morphology of the Al metallization layer after 2000 s of power cycling calculated using the three models. Figure 18 shows the variation in *S*_a_ of the three grain models with average grain sizes of 4 μm, 5 μm, and 6 μm with the power cycling time. The model with an average grain size of 4 μm has the lowest *S*_a_, whereas the models with average grain sizes of 5 μm and 6 μm have higher and similar *S*_a_s. After 2000 s of power cycling, the *S*_a_ of the three models with average grain sizes of 4 μm, 5 μm, and 6 μm is 0.100 μm, 0.119 μm, and 0.123 μm, respectively. That is, the larger the grain size, the rougher the surface of the Al metallization layer. This result is obtained because the number of grain boundaries per area of the Al metallization layer decreases as the grain size increases. Consequently, there are fewer constraints on the slip plane in the grains during slipping, and the grains deform more severely, leading to a significant increase in the surface roughness.

The *S*_a_s of the models with average grain sizes of 5 μm and 6 μm become similar as the power cycling time increases. The model with an average grain size of 5 μm has a slightly higher *S*_a_ than the model with an average grain size of 6 μm before 1560 s. After 1560 s, the *S*_a_ of the model with an average grain size of 6 μm surpasses that of the model with an average grain size of 5 μm, and the difference between the *S*_a_s of the two models trends upward. This result is obtained due to the disparity in the average grain orientation differences of the two models. As there are few grain boundaries, it is difficult to maintain consistency between the average grain orientation differences of the two models. Here, the models with average grain sizes of 5 μm and 6 μm have average grain orientation differences of 44.628° and 40.324°, respectively, i.e., the model with an average grain size of 6 μm has a smaller orientation difference than the model with an average grain size of 5 μm. The results presented in Section 4.3 show that a large grain orientation difference can increase the surface roughness. Therefore, at the initial stage, the *S*_a_ of the model with an average grain size of 5 μm increases faster, as its orientation difference is greater than that of the model with an average grain size of 6 μm. Figure 17c shows that the strain is all concentrated in a small region because the small average grain orientation difference results in a lower intergranular strain in the models with larger grain sizes. However, after a certain period of power cycling, the *S*_a_ of the model with an average grain size of 6 μm is higher than that of the model with an average grain size of 5 μm. This result indicates that large grains have a stronger effect on the surface roughness than small grains.

In addition, under the condition of fewer cycles, the intercrystal shear stress is small, and some grains do not reach the critical shear stress required for sliding. The smaller the orientation difference, the less easy it is to slip. The model with an average grain size of 5 μm has more grain deformation and slip, which results in greater surface roughness changes. Therefore, at the initial stage, the *S*_a_ of the model with an average grain size of 5 μm grows faster, since its orientation difference is greater than that of the model with an average grain size of 6 μm. However, with the increase in power cycles, the intercrystallite stresses of most grains satisfy grain slip conditions, and the effect of grain size on surface roughness becomes obvious. Then, the *S*_a_ value of the model with an average grain size of 6 μm is higher than that of the model with an average grain size of 5 μm.

Figure 19 presents the experimentally determined cross-section of the Al metallization layer at the IGBT chip center at the initial state and after 100 k cycles. This result indicates that the microstructure of the Al metallization layer evolves during power cycling. In the initial state, the Al grains appear as columns with a height equal to the layer thickness (Figure 19a). The average grain size is 6.356 μm. After power cycling, the Al grains are refined (Figure 19b), and the average grain size is 4.090 μm. During power cycling, small grains can slow down the roughening process of the Al metallization layer.

## 5. Conclusions

The aim of this study was to investigate the major factors affecting the morphology evolution of the Al metallization layer of an IGBT chip during power cycling. Here, a DC power cycling test is performed on an IGBT module, the evolution of the microstructure of the Al metallization layer is observed from the initial stage to failure using SEM, AFM, FIB, and EBSD, and the *S*_a_ of the layer is measured quantitatively. During power cycling, Al grains undergo plastic deformation, whereby the Al metallization layer rapidly undergoes morphological changes from flat to coarse. The surface of the Al metallization layer on the IGBT chip has an uneven distribution. The surface roughness is considerably larger in the central area of the chip than in the edge area.

A thermal–electrical–structural analysis is conducted on IGBT modules under power cycling test conditions. Multilevel submodeling technology and the Voronoi algorithm are employed to generate a polygonal grain geometric model that captures the geometric characteristics of the Al grains. Numerical simulation and the analysis of the evolution of the surface roughness of the Al metallization layer are performed using a crystal thermo–elasto–plasticity constitutive equation combined with an ABAQUS FE analysis through the user-defined subroutine UMAT.

The effects of the temperature, grain orientation, and grain size on the evolution of the surface roughness of the Al metallization layer during power cycling are also analyzed. The temperature is the main external influence factor for the surface roughness. A high temperature leads to severe degradation of the surface morphology of the Al metallization layer, and increases the surface roughness. As the number of power cycles increases and plastic deformation accumulates, the rate of change in the surface roughness gradually increases. Internal factors, including the grain size and grain orientation, also have a significant effect on the surface roughness. During power cycling, the large orientation difference between adjacent grains results in a large difference in the deformation between adjacent grains, increasing the roughness of the deformed surface. Large grains correspond to few grain boundaries per area for the Al metallization layer. Thus, the slip planes inside the Al grains are less constrained by the grain boundaries during the slip process, and the grains are prone to deformation, resulting in an increase in the surface roughness. The degradation of the Al metallization layer can be mitigated by lowering the surface temperature of the IGBT chip, and by reducing the grain orientation difference and grain size of the Al metallization layer.

In closing, it is necessary to mention that this study is limited to thermo–elasto–plastic simulation at low temperatures, and the effect of thermodynamic coupling is ignored here. However, the coupling effect of the thermal strain and elastic strain tensor is significant at high temperatures. In addition, during the power cycling test, subgrain boundaries will gradually form, so there will be subgrains within large grains, i.e., there is a grain refinement phenomenon. Grain refinement will hinder the surface roughness change to a certain extent. The next goal is to evaluate the surface roughness evolution while considering the effect of thermodynamic coupling and grain refinement in future work.

## Figures and Tables

**Figure 1 materials-16-01936-f001:**
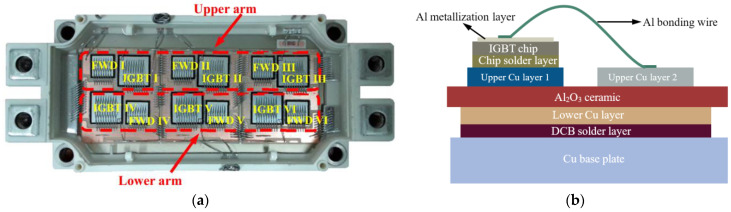
Test IGBT module. (**a**) View of the tested IGBT module; (**b**) Schematic cross-section of the module.

**Figure 2 materials-16-01936-f002:**
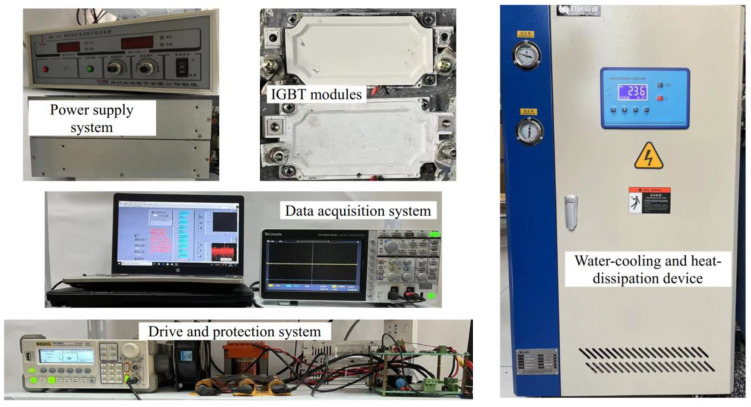
Power cycling test platform.

**Figure 3 materials-16-01936-f003:**
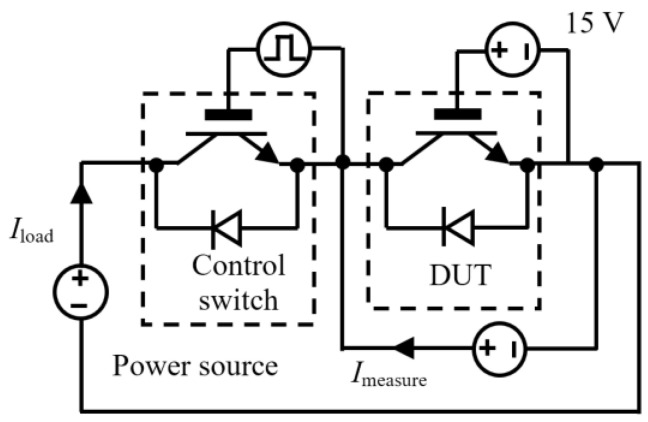
Circuit diagram of the power cycling test platform.

**Figure 4 materials-16-01936-f004:**
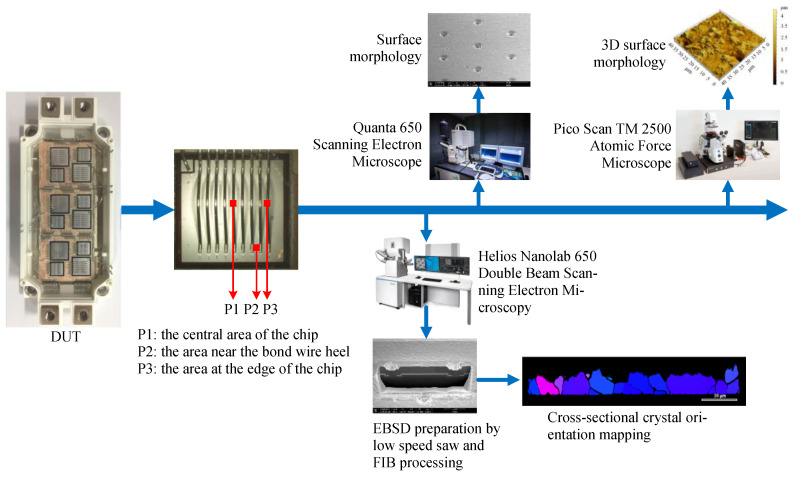
Experimental process for observing the microstructure of the Al metallization layer.

**Figure 5 materials-16-01936-f005:**
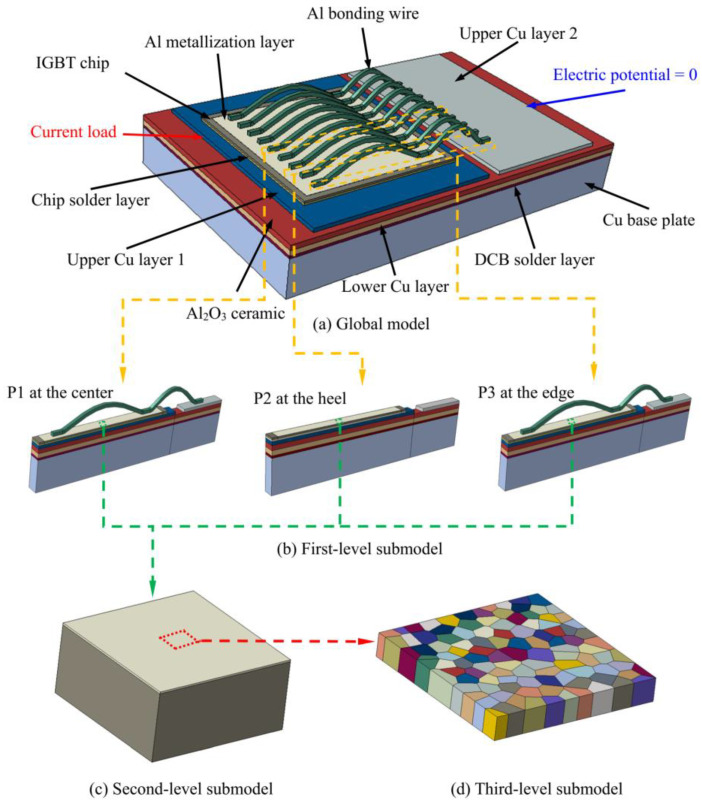
Geometric model for the global model and submodel of the IGBT module.

**Figure 6 materials-16-01936-f006:**
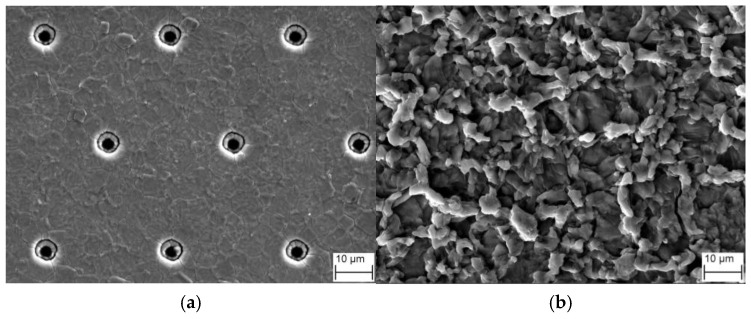
SEM images of the surface morphology of the Al metallization layer during power cycling: (**a**) at zero cycles and (**b**) after 193 k cycles.

**Figure 7 materials-16-01936-f007:**
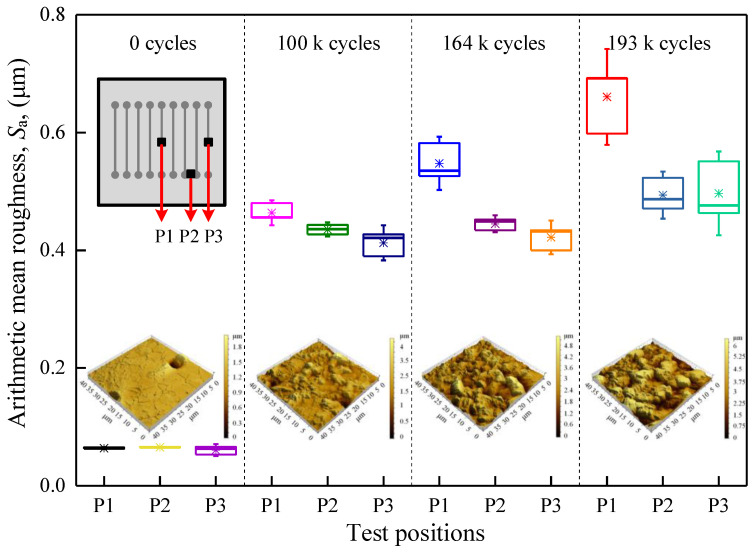
Distribution of the arithmetic mean surface roughness of the Al metallization layer of the IGBT chip during power cycling. The “*” in the figure is the mean point.

**Figure 8 materials-16-01936-f008:**
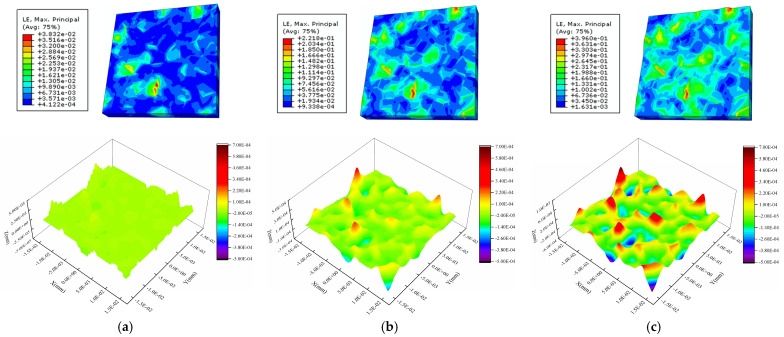
Strain and height profiles for the Al metallization layer surface at the central chip area (P1) after different power cycle times: (**a**) 200 s, (**b**) 1200 s, and (**c**) 2000 s.

**Figure 9 materials-16-01936-f009:**
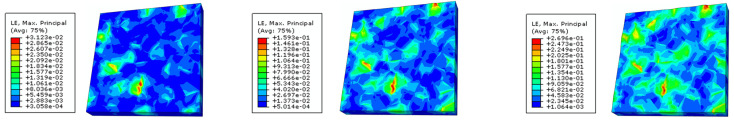
Strain and height profiles for the surface of the Al metallization layer at the bond wire heel (P2) after different power cycle times: (**a**) 200 s, (**b**) 1200 s, and (**c**) 2000 s.

**Figure 10 materials-16-01936-f010:**
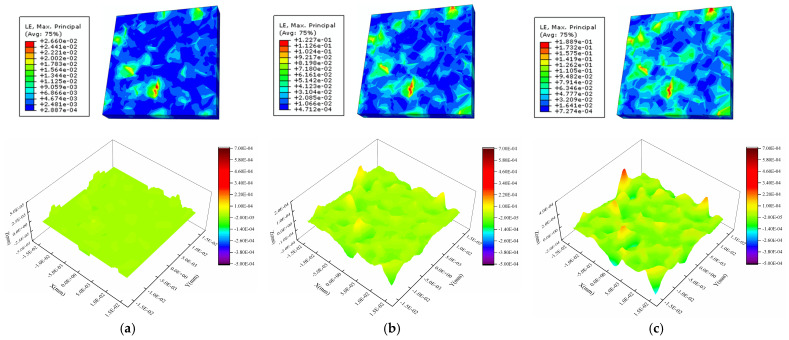
Strain and height profiles for the surface of the Al metallization layer at the chip edge (P3) after different power cycle times: (**a**) 200 s, (**b**) 1200 s, and (**c**) 2000 s.

**Figure 11 materials-16-01936-f011:**
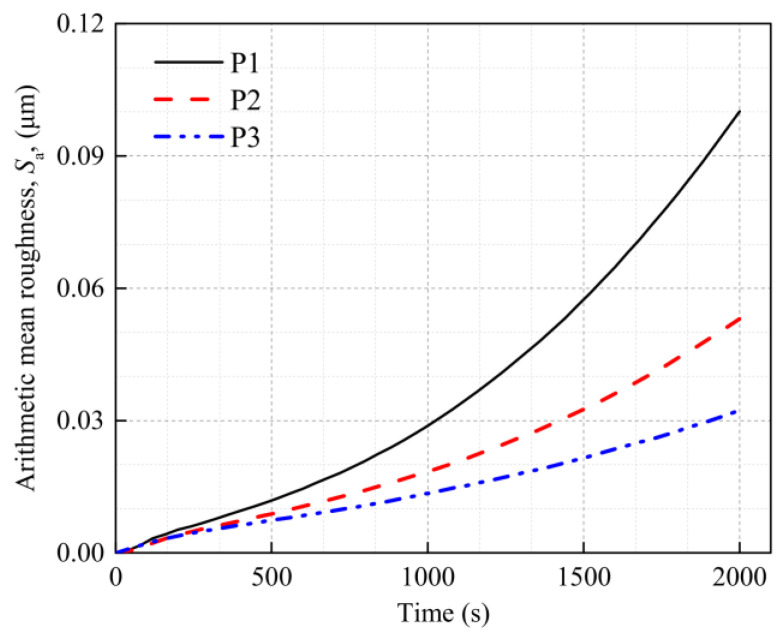
Evolution of the arithmetic mean surface roughness of the Al metallization layer at different test positions during power cycling.

**Figure 12 materials-16-01936-f012:**
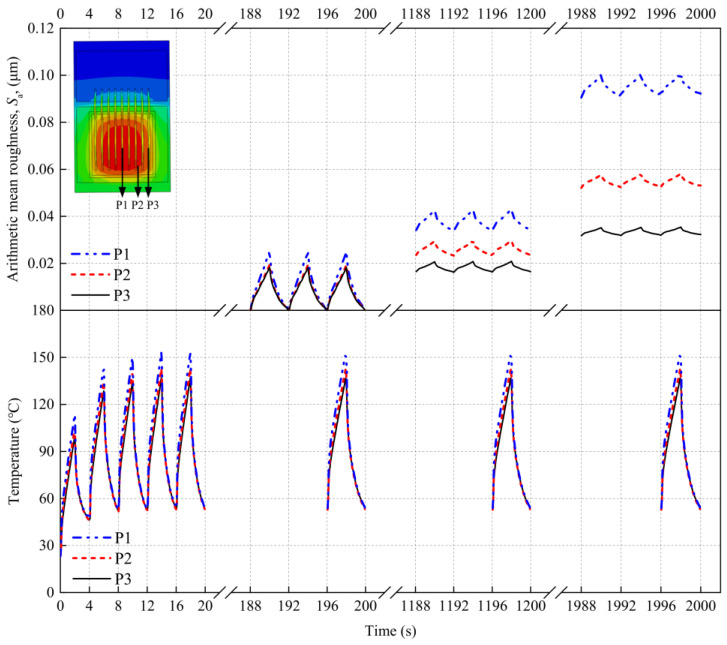
Variation in the arithmetic mean surface roughness of the Al metallization layer at different power cycling times in the presence of temperature fluctuations.

**Figure 13 materials-16-01936-f013:**
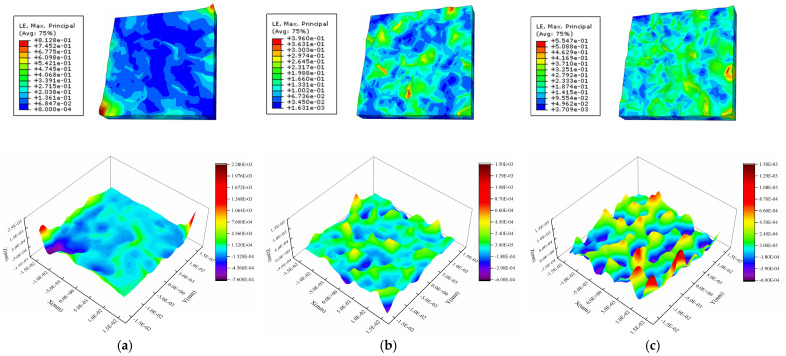
Strain and height profiles for the Al metallization layer after 2000 s of power cycling with various mean grain orientation differences: (**a**) 15°, (**b**) 45°, and (**c**) 90°.

**Figure 14 materials-16-01936-f014:**
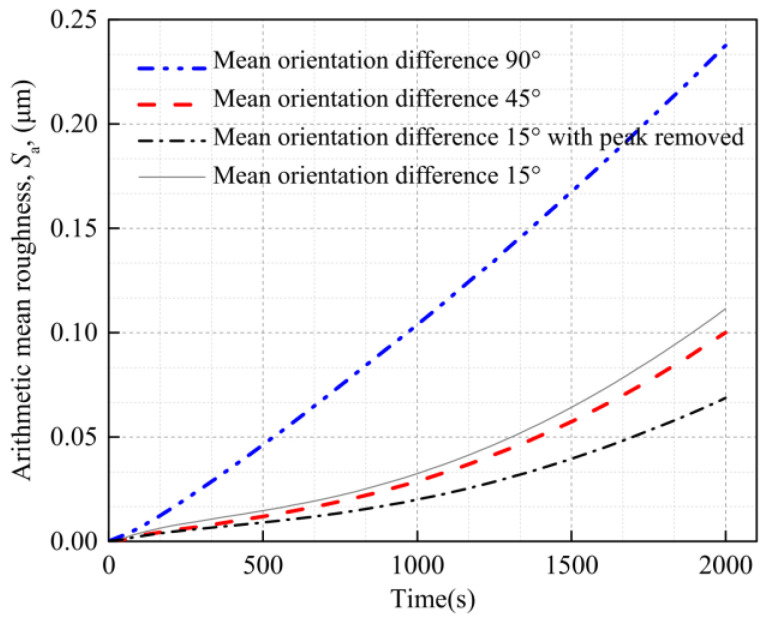
Effect of the orientation difference on the surface roughness of the Al metallization layer.

**Figure 15 materials-16-01936-f015:**
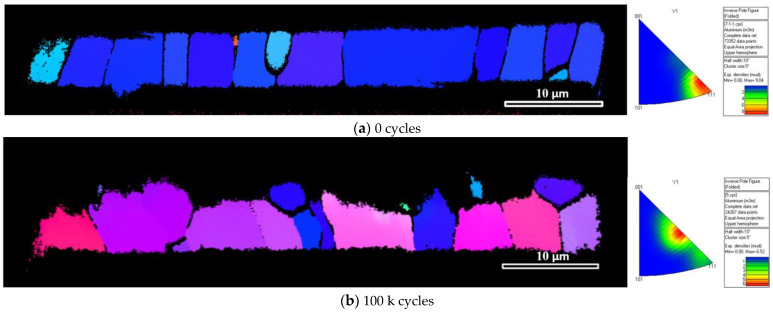
EBSD grain orientation maps and inverse pole figures of the Al metallization cross-section at the chip center: (**a**) in the initial state and (**b**) after 100 k cycles.

**Figure 16 materials-16-01936-f016:**
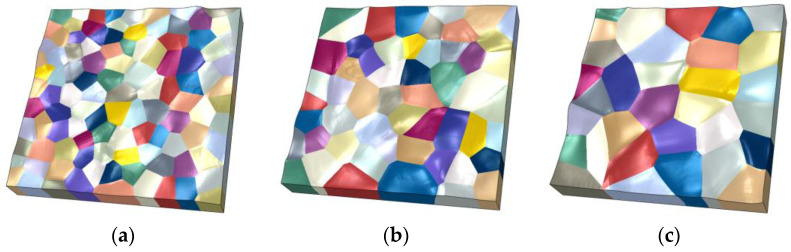
Models for the Al metallization layer with different grain sizes. (**a**) 4 μm; (**b**) 5 μm; (**c**) 6 μm.

**Figure 17 materials-16-01936-f017:**
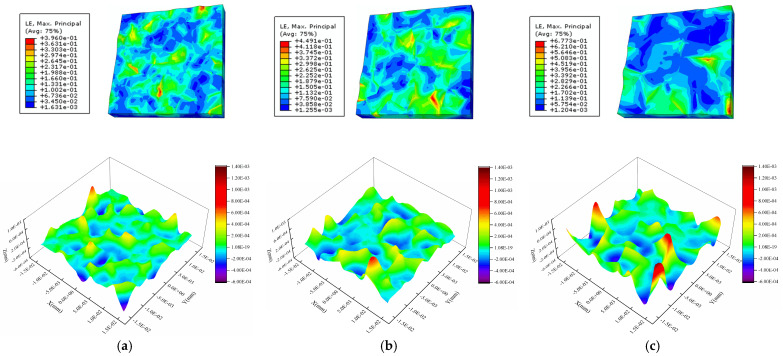
Strain and height profiles after 2000 s of power cycling for the Al metallization with different grain sizes: (**a**) 4 μm, (**b**) 5 μm, and (**c**) 6 μm.

**Figure 18 materials-16-01936-f018:**
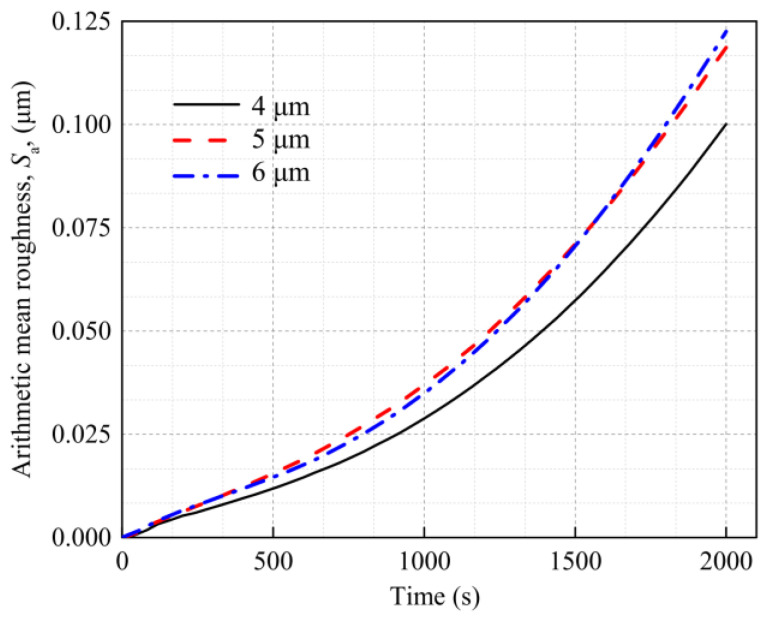
Effect of the grain size on the surface roughness of the Al metallization layer.

**Figure 19 materials-16-01936-f019:**
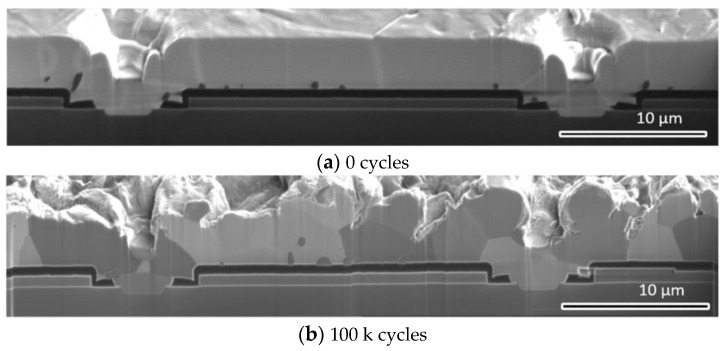
Cross-section of the Al metallization layer at the chip center: (**a**) initial state and (**b**) after 100 k cycles.

**Table 1 materials-16-01936-t001:** Geometric dimensions of the global model of the IGBT module.

IGBT Layer, Measure	Length (mm)	Width (mm)	Thickness (mm)
IGBT chip	13	13	0.15
IGBT chip solder	13	13	0.15
Upper copper layer 1	15	18	0.30
Upper copper layer 2	10	18	0.30
Ceramic substrate	19	30	0.40
Lower copper layer	19	30	0.40
Base solder	98	44	0.20
Copper base plate	120	60	3.00

**Table 2 materials-16-01936-t002:** Material parameters of the crystal plasticity model of the Al metallization layer.

Material Parameters		Symbol	Value	Unit
Elastic parameters	Elastic constants	*C* _11_	108,000	MPa
		*C* _12_	62,000	MPa
		*C* _44_	28,300	MPa
Plastic parameters	Rate-dependent sensitivity exponent	*n*	10	-
	Reference shear strain rate	${\dot{\gamma }}_{0}$	0.001	s^−1^
	Initial hardening modulus	*h* _0_	60	-
	Saturation stress	*τ_s_*	61	MPa
	Initial critical resolved shear stress	*τ* _0_	21.17	MPa
	Ratio of latent hardening to self-hardening	*q*	1.4	-
	Volume thermal expansion coefficient	*A*	24	ppm/°C

**Table 3 materials-16-01936-t003:** Material properties applied in the electro–thermo–structural FE analysis.

Materials	Young’s Modulus (GPa)	Poisson’s Ratio	Coefficient of Thermal Expansion (ppm/°C)	Thermal Conduction (W/m·K)	Electrical Conductivity (1/mΩ·mm)	Density (kg/m^3^)	Specific Heat (J/kg·K)
Cu [32]	128	0.36	16.4	400	59.523	8920	380
Ceramic [32]	345	0.25	7.2	20	1 × 10^−18^	3960	753
Al [32]	70.6	0.34	24	237	37.735	2700	900
Si [32]	130	0.22	2.5	148	Temperature-dependent	2330	700
Sn3.0Ag0.5Cu solder [33]	10.6	0.35	25	57	9.615	7300	230

## Data Availability

Data is contained within the article.

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
