# Peer review of "Macro–Mesoscale Modeling of the Evolution of the Surface Roughness of the Al Metallization Layer of an IGBT Module during Power Cycling"

_materials, 2023, doi:10.3390/ma16051936_

Round 1

Reviewer 1 Report

The aim of the reviewed article is important. The presented results of investigations are interesting, but the description of performed research needs improving. The article can be accepted for publication after revision.

In the revised version of this article the Authors should take into account following remarks:

1. The schematic diagram of the tested IGBT module should be shown.

2. The diagram shown in Figure 3 are wrong. In my opinion the voltage sources should be connected to the gates instead of collectors of the considered transistors or the used symbols of IGBTs are wrong.

3. In the used model the equations describing electrical and thermal properties of the semiconductor devices are not given. It is not clear for me how these properties are described.

4. It is not clear for me how temperature waveform shown in Fig. 12 was obtained. To which point of the module corresponds this temperature?

5. The reference list should be extended. Some papers describing electrical and thermal properties of IGBT modules should be included in this list with adequate citations in the body of the paper. Such problem is described e.g. in the articles by L. Codecasa, V. d'Alessandro, J. Zarębski or D. Wojciechowski.

Reviewer 2 Report

1.       The paper has many grammatical points. It should be evaluated by a native person.

2.       Uncertainty analysis of the experimental procedure is necessary.

3.       It would be very useful if the authors added some more detail about the development of the UMAT subroutine in the paper.

4.       In figure 7, which reason causes to increase of the roughness with the power cycle? It should be explained in more detail.

5.       In figures 8 to 10 some numbers can not be seen conveniently.

6.       Why the experimental and numerical results were not compared? It is mandatory for the validation of results.

7.       Why does figure 18 the grain size for 5 μm behave reversely before 1500 sec than the other 2 cases? In the other words, why this case increases roughness before 1500 sec more than 6 μm?

8.       The future work plan should be clarified at the end of the conclusion.

9.       The literature review is not completely good. Then, the novelty of the work is in doubt. Some new works should be considered. Also, it is recommended to cite this work in your revised version:

·         Valizadeh Yaghmourali, Yousef, Nima Ahmadi, and Ebrahim Abbaspour-sani. "A thermal-calorimetric gas flow meter with improved isolating feature." Microsystem Technologies 23, no. 6 (2017): 1927-1936.

Round 2

Reviewer 1 Report

The Authors took into account only some of my remarks given in the previous reports. In my opinion, the second revision should be performed. In the revised version of this article following remarks should be taken into account.

1. Detailed description of Figure 3 should be added. All components visible in this figure should be described.

2. Information about used description of electrical and thermal properties of the modelled device should be given in the article.

3. Some papers describing electrical and thermal properties of IGBT modules should be cited and some information about modelling such modules using compact electrothermal models should be added. There are many articles published in journals e.g. by IEEE or MDPI describing such models.

Reviewer 2 Report

The paper is acceptable now.

Author Response

Thanks for the constructive comments, and it is really useful for our work.